# Human RTEL1 Interacts with KPNB1 (Importin β) and NUP153 and Connects Nuclear Import to Nuclear Envelope Stability in S-Phase

**DOI:** 10.3390/cells12242798

**Published:** 2023-12-08

**Authors:** Michael Schertzer, Laurent Jullien, André L. Pinto, Rodrigo T. Calado, Patrick Revy, Arturo Londoño-Vallejo

**Affiliations:** 1Institut Curie, PSL Research University, CNRS, UMR3244, F-75005 Paris, France; michael.schertzer@curie.fr; 2Sorbonne Universités, CNRS, UMR3244, F-75005 Paris, France; 3INSERM UMR 1163, Laboratory of Genome Dynamics in the Immune System, Equipe Labellisée Ligue Contre le Cancer, F-75006 Paris, France; laurent.jullien@ligue-cancer.net (L.J.); patrick.revy@inserm.fr (P.R.); 4Paris Descartes–Sorbonne Paris Cité University, Imagine Institute, F-75015 Paris, France; 5Department of Medical Imaging, Hematology, and Oncology, Ribeirao Preto Medical School, University of Sao Paulo, Ribeirao Preto 14049-900, Brazil; alpintosantos@gmail.com (A.L.P.); rtcalado@fmrp.usp.br (R.T.C.)

**Keywords:** Regulator of TElomere Length Helicase 1, protein import, nuclear envelope, nuclear pore, KPNB1 (importin β), NUP153, S-phase

## Abstract

Regulator of TElomere Length Helicase 1 (RTEL1) is a helicase required for telomere maintenance and genome replication and repair. RTEL1 has been previously shown to participate in the nuclear export of small nuclear RNAs. Here we show that RTEL1 deficiency leads to a nuclear envelope destabilization exclusively in cells entering S-phase and in direct connection to origin firing. We discovered that inhibiting protein import also leads to similar, albeit non-cell cycle-related, nuclear envelope disruptions. Remarkably, overexpression of wild-type RTEL1, or of its C-terminal part lacking the helicase domain, protects cells against nuclear envelope anomalies mediated by protein import inhibition. We identified distinct domains in the C-terminus of RTEL1 essential for the interaction with KPNB1 (importin β) and NUP153, respectively, and we demonstrated that, on its own, the latter domain can promote the dynamic nuclear internalization of peptides that freely diffuse through the nuclear pore. Consistent with putative functions exerted in protein import, RTEL1 can be visualized on both sides of the nuclear pore using high-resolution microscopy. In all, our work points to an unanticipated, helicase-independent, role of RTEL1 in connecting both nucleocytoplasmic trafficking and nuclear envelope integrity to genome replication initiation in S-phase.

## 1. Introduction

Regulator of TElomere Length Helicase 1 (RTEL1) is an essential protein, required for telomere length maintenance [1,2,3,4,5,6], DNA repair [7,8,9] and genome replication [7,10,11,12]. Germline variants in the RTEL1 gene are etiologic in telomere biology disorders (TBD) [13,14,15,16], especially regenerative-defective disorders, such as dyskeratosis congenita (DC) and its more severe presentation, Høyeraal-Hreidarsson Syndrome (HHS), but also, in the heterozygous state, with lung fibrosis, bone marrow failure and other clinical manifestations [17,18,19,20,21,22]. The protein has been proposed to act as an ATP-dependent helicase at telomeres allowing progression of the replication fork beyond the T-loop or the extension of the G-rich overhang by telomerase [2,3,5]. Elsewhere in the genome, the protein may act to resolve obstacles posed to the replication fork such as G-quadruplexes or RNA:DNA hybrids [11,12]. We have also demonstrated, in a previous study, the role of human RTEL1 in nucleocytoplasmic transport (NCT), specifically the export of the snRNA pre-U2 from the nucleus to the cytoplasm in association with XPO1, a major nuclear export receptor [23]. 

In this study, we have set up experiments to further explore the potential roles of RTEL1 in nucleocytoplasmic exchanges, specifically its interactions with major nuclear import receptors such as importin α (official name KPNA2) and importin β (official name KPNB1) [24,25,26], and explored how these roles may be connected to nuclear envelope deformations, which appear to be prevalent in HHS patients’ cells (this study). 

## 2. Materials and Methods

### 2.1. Cell lines and Culture Conditions

HeLa cell lines and transformed HHS patient cells were cultured in Dulbecco’s modified Eagle’s medium supplemented with L-glutamine, sodium pyruvate, non-essential amino acids (Life Technologies, Carlsbad, CA, USA) and 10% fetal calf serum (Bio-West, Riverside, CA, USA). HeLa cells are from the Institute’s collection. HeLa cells stably expressing the Tet repressor were from Life Technologies (T-REX tetracycline system). Construction of HeLa cells inducible for RTEL1 expression has already been described [23]. Primary skin fibroblasts were obtained from healthy volunteers and from DC patients in the Imagine Institute (Paris) and the bone marrow failure clinic, University Hospital, University of Sao Paulo (Riberao Preto, Brazil), respectively. Clinical features of French patients have been described in for P-2 and P-5 in [27]. Clinical features for one Brazilian patient will be described separately. The DC patient carried biallelic RTEL1 mutations (Y1086C/A1259Lfs2*). As described before [23], fibroblasts were obtained from skin biopsies and maintained as primary cultures under 4% 02/5%CO_2_ or maintained as cell SV40ER/TERT transformed/immortalized cell lines. 

### 2.2. Constructs

For inducible expression, the T-REX tetracycline system (Life Technologies, Carlsbad, CA, USA) was used. The constructs were cloned as EcoRI/XhoI fragments into pcDNA-4/TO as previously described [23].

Constructs carrying full-length wild-type RTEL1, or the versions carrying mutations in the RING (MR:C1265A, C1268A) or in the Nuclear Localization Signal (NLSmut), have been previously described [23]. Detailed descriptions of all constructs used in this study are included in a Appendix A. A diagram is presented in Figure 1. 

RTEL1 CRISPR plasmid pools and control (SCR) plasmids were from Santa Cruz (sc-403196A1, and sc-418922, respectively). All sequencing was performed in-house at the Institut Curie Sequencing center using Big Dye Terminator 3.1 (Thermofisher, Waltham, MA, USA). Oligonucleotides for sequencing and PCR were synthesized by Life Technologies.

### 2.3. Transfections

Plasmid transfections were carried out using Effectene (Qiagen, Hilden, Germany) according to the manufacturer’s instructions. CRISPR knockdown transfections were carried out using UltraCruz transfection reagent (Santa Cruz Biotechnologies, Dallas, TX, USA).

### 2.4. Drug Treatments

Importazole (Sigma-Aldrich, St. Louis, MI, USA) was used at 40 μg/mL concentration for 3 h, Ivermectin (Sigma-Aldrich) at 50 μM for 3 h, Aphidicolin (Sigma-Aldrich) at 1 μg/mL for 1 h and the CDC7 inhibitor PHA76749 (Sigma-Aldrich) 5 μM for 1 h. Synchronization of cells at G1/S was accomplished via a 2×Thymidine block (2 mM thymidine (Sigma) for 18 h, release for 9 h, then thymidine for 18 h, during which time cells were induced for transgenes with tetracycline 1 μg/mL (Sigma).

### 2.5. Anti-hRTEL1 Antibody Production

The RTEL1 polyclonal antibody was prepared as described [23]. 

### 2.6. Commercial Antibodies

A list of all commercial antibodies used in this study is included in Appendix A.

### 2.7. Immunofluorescence

Cells grown on slides with 50.000 cells were fixed in 3% formaldehyde, 1X PBS, 300 mM sucrose for 15 min, and permeabilized in 0.5% Triton-X-100, 20 mM Tris-HCl pH 8.0, 50 mM NaCl, 5 mM MgCl_2_ and 300 mM sucrose. Fixed cells were blocked with 10% horse serum in PBS and incubated sequentially in different primary antibodies, followed by fluorescently labeled secondary antibodies. All incubation steps were performed in a humid incubator at 37 °C for 1 h. Slides were mounted in Vectashield with 0.2 μg/mL 4_, 6-diamidino-2-phenylindole (DAPI). Images were taken with a 3D deconvolution microscope (Leica DM6000 B or Nikon) (Leica, Wetzlar, Germany; Nikon, Tokyo, Japan) using the MetaMorph v7.8 software. Final images are composed of arithmetic stacks of 20–30 deconvolved images, each 0.2 μm in depth. For SIM microscopy, the Nikon N-SIM-S was used on cells as described above and Prolong Gold (Thermofisher) was used as an anti-fade without DAPI.

### 2.8. Circularity

Measures of nuclear circularity were carried out with the ImageJ 1.54g plug-in Circularity, where circularity = 4π (area/perimeter^2^). 

### 2.9. Western Blot Analyses

Proteins were separated in 4–12% sodium dodecylsulfate-polyacrylamide gel electrophoresis (SDS PAGE) gels (Life Technologies) and then transferred to PVDF membranes (GE Healthcare, Chicago, IL, USA) for immunodetection. HRP-linked secondary antibodies (DAKO) were revealed using chemiluminescent detection (ECL 2; GE Healthcare) on the Chemidoc system (Bio-Rad, Hercules, CA, USA). 

### 2.10. Immunoprecipitations

Lysates were prepared using NP40 lysis buffer (50 mM Tris–HCl pH 6.8, 150 mM NaCl, 1% NP-40,) plus EDTA-free protease inhibitor cocktail (Roche, Basel, Switzerland) and Halt Phosphatase Inhibitor (Thermofisher). Lysates were treated with Benzonase (Sigma-Aldrich) for 45 min at RT and then cleared by centrifugation. Approximately 500 μg of lysates were incubated with 10 μg of antibody or nonspecific IgG. Complexes were bound to protein A or G agarose beads (Millipore, Burlington, MA, USA). Beads were washed 3× with lysis buffer and 1× with phosphate-buffered saline (PBS). Beads were resuspended in sample buffer (0.125MTris–HCl, 4% SDS, 20% glycerol, 10% 2-mercaptoethanol, 0.004% bromophenol blue) and boiled. Proteins were analyzed via Western blot.

### 2.11. Generation of CRISPR/Cas9 RTEL1 Knockout

HeLa cells were transfected with RTEL1 CRISPR/Cas9 KO Plasmid (h) (Santa Cruz, sc-403196) or a control CRISPR/Cas9 Plasmid (Santa Cruz, sc-418922). The RTEL1 CRISPR/Cas9 KO Plasmid targets Exon 6. Plasmids express a GFP, which allows enrichment of highly transfected cells via cell sorting. GFP-positive cell populations were put back into culture and used for experiments (IFs, Western blots and rescue) within 24 h.

### 2.12. Statistical Analysis

Measures of circularity and percentages of nuclear/cytoplasmic distributions were compared using a non-parametric test (Mann–Whitney). The number of cells examined for each experiment is indicated below every violin plot. 

## 3. Results

### 3.1. RTEL1 Deficiency Is Associated with Nuclear Envelope Deformations That Specifically Occur in S-Phase

While characterizing immortalized fibroblast cells from HHS patients carrying RTEL1 mutations, we observed cells displaying pronounced nuclear envelope (NE) deformations, with obvious loss of nuclear circularity (Appendix A). The NE of these cells is characterized by a heterogenous staining with antibodies against LMNB1 and LMNA/C and the presence of blebs depleted for LMNB1 (Appendix A), even though LMNB1 levels, as detected by Western blot, are similar to those in wild-type cells (Appendix A). These deformations were also present in primary dermal fibroblasts obtained from a DC patient carrying RTEL1 mutations, thus indicating that this phenotype was independent from both the immortalization status of the RTEL1-mutated cells and the severity of the disease (Appendix A). To demonstrate RTEL1-dependence of such abnormalities, we designed experiments to deplete RTEL1 from HeLa cells using CRISPR technology. Protein depletion (Appendix A) was associated with similar NE defects and loss of circularity (Figure 2A,B), strongly suggesting that the phenotype is the direct consequence of the loss of RTEL1 function. This conclusion was further supported by the fact that the NE abnormalities observed in RTEL1-depleted cells were fully complemented by the ectopic expression of WT RTEL1 (Figure 2C,D), which excluded the possibility of a CRISPR/Cas9 off-target effect. Interestingly, similar NE instabilities could also be provoked by the controled inducible expression of exogenous RTEL1 carrying engineered mutations that mimic those of HHS patients (Figure 2E,F), in this case mutations affecting the RING domain (hereafter named RTEL1-MR, for Mutated Ring). 

During these experiments, we observed that not all cells expressing the mutated protein displayed NE instability, which suggested possible cell cycle dependence. Indeed, in both HHS patient cells and in HeLa cells expressing RTEL1-MR, nuclear deformations were almost exclusively observed in cells that were in S-phase, as indicated by positive staining for EdU (Appendix A). To confirm this observation, we used our inducible system in HeLa cells to express exogenous RTEL1 (wild-type or MR) in cells blocked in G1/S via a double thymidine treatment and then released into S-phase (Appendix A). Cells blocked in G1/S showed no NE deformations (Appendix A); however, as soon as 1 h after release into S-phase, cells expressing exogenous RTEL1-MR displayed abnormally shaped nuclei (Appendix A). 

Because RTEL1 has been shown to interact directly with PCNA [6], we considered the possibility that a defect in RTEL1 function during genome DNA replication might be responsible for the NE instability. However, nuclear deformations were also detected in cells released into S-phase in the presence of aphidicolin at doses that inhibit all polymerases [28] (Figure 1G), suggesting that it is not at the level of DNA synthesis where RTEL1 should act to preserve NE integrity. To determine whether replication origin firing in S-phase was required for these NE deformations to appear in the presence of *RTEL1* mutations, we released RTEL1-MR expressing cells into S-phase in the presence of a CDC7 inhibitor, a major kinase whose activity is required to trigger origin firing [29]. Outstandingly, inhibition of origin firing was sufficient to rescue the NE instability in these cells (Figure 2H,I), revealing a direct link between NE stability, RTEL1-dependent functions and the initiation of genome replication in S-phase.

### 3.2. RTEL1 Is Imported to the Nucleus When Cells Enter S-Phase 

In a previous study, we showed that RTEL1 is present in both the cytoplasm and the nucleus [23]. To explore whether there are any dynamic changes in the RTEL1 cytoplasmic/nuclear distribution in relation to S-phase, we measured, using immunofluorescence, the relative cytoplasmic and nuclear levels of endogenous RTEL1 in HeLa cells, both blocked at G1/S and released for 1 h into S-phase. As illustrated in Figure 3A,B, endogenous RTEL1 shows a preferential nuclear localization (slightly above 60% following this approach) in G1/S, which significantly increases (reaching almost 80%) upon entry into S-phase. This result indicates a cell cycle-specific behavior for this protein that could be potentially linked to the NE phenotype described above. To test whether RTEL1 import in S-phase depended on active protein import, we treated cells with importazole, a specific inhibitor of KPNB1 [30]. Indeed, importazole treatment prevented the S-phase-associated enrichment of RTEL1 in the nucleus (Figure 3B), indicating an active process. 

To further explore the connections between protein import, S-phase-associated NE defects and RTEL1, we resorted to our experimental system described above. When the wild-type RTEL1 protein was overexpressed, it showed a strong nuclear enrichment (>80%) in both G1/S-blocked cells and cells released into S-phase (Figure 3C,D). On the other hand, the overexpressed RTEL1-MR showed a lower level of nuclear enrichment (slightly above 70%) in G1/S cells, which again did not change when cells were released into S-phase (Figure 3C,D). This result points to a potential role of the RING domain in the nuclear enrichment of the protein. To further explore both how nuclear enrichment is connected to NE defects and how much the nuclear localization of RTEL1 depends on the classic import pathway, we introduced mutations in the canonical nuclear localization signal (NLS) [31] of the protein found between the helicase and the C-terminus domain (positions 872-880) (Figure 1). As shown in Figure 3C,D, the exogenous NLS-mutated version of RTEL1 accumulated significantly less in the nucleus compared to the WT or to the RTEL1-MR protein, but its localization was still essentially nuclear (slightly above 60%), demonstrating that RTEL1 can be imported through a KPNA2-independent mechanism. Nevertheless, the NLS-mutated protein (hereafter simply noted NLSmut) did not show any variation when cells entered S-phase, and its expression was associated with a modest loss of nuclear circularity (Figure 3E, Appendix A) that contrasted with the very pronounced phenotype following the induction of RTEL1-MR expression (Appendix A, Figure 3E, Appendix A). Also, in contrast with the latter is the fact that the phenotype induced by expression of NLSmut is present in cells both before and after their release in S-phase. Taken together, these results suggest that the reason RTEL1 MR induces severe NE instability specifically in S-phase is likely unrelated to the lack of nuclear enrichment of this protein at that point of the cell cycle.

### 3.3. RTEL1 Overexpression Protects Cells from NE Defects Induced by Inhibition of KPNB1

The use of importazole to explore the dependence of RTEL1 on KPNB1 led to an unexpected observation: treatments of HeLa cells with this inhibitor induced conspicuous NE defects and loss of nuclear circularity in G1/S-blocked cells (Figure 4A,B). This phenotype was very similar to that observed in S-phase cells expressing mutated RTEL1 in regard to lamin composition (Figure 4A). To determine whether there was any connection between these importazole-induced NE defects and RTEL1 itself, cells that overexpressed exogenous RTEL1 were treated with importazole. Importazole had a limited, but significant, impact on the nuclear localization of the overexpressed WT protein (Appendix A). Strikingly, cells did not show any NE defect or loss of circularity (Figure 4A,B), indicating that overexpressed RTEL1 was somehow able to protect against importazole-induced NE defects. In contrast, RTEL1-MR, which also accumulated less in the nucleus in the presence of the inhibitor (Appendix A), was unable to protect cells against importazole (Figure 4A,B). This result suggests that RTEL1 exerts its NE stabilization effect in the presence of importazole through some specific, likely RING-related, function and not just from being expressed in excessive amounts in these cells. As expected, the importazole treatment did not further impact the cytoplasmic/nuclear distribution of the version of RTEL1 carrying the NLS mutations (Appendix A), indicating an epistatic relationship between the inhibition of KPNB1 and inhibition of the classic import pathway. Remarkably, this version of RTEL1 almost completely protected the cells from the extensive NE defects induced by importazole (Figure 4A,B), revealing that RTEL1 is able to exert this protection even when it is unable to use the classical import pathway.

Thus far, our experiments point to an unanticipated role for RTEL1 in nuclear envelope stabilization, and strongly suggest that this role might be tightly connected to an KPNB1-dependent import pathway but not to the NLS-mediated classic import pathway that involves both KPNB1 and KPNA2 [24]. Interestingly, the nuclear localization of RTEL1-MR appeared to be extremely sensitive to ivermectin (IVM, a specific inhibitor of KPNA2 [32]) before and after entry into S-phase, while the same treatment had a limited impact on WT RTEL1 localization (Figure 4C, Appendix A), suggesting an exquisite dependence of MR on KPNA2. This result both definitely establishes that an intact RING domain is required for RTEL1 to be enriched in the nucleus and reinforces the conclusion that, while WT RTEL1 is able to act upon (and likely use) a specific KPNB1-dependent pathway, RTEL1-MR is not.

### 3.4. RTEL1 Interacts, Albeit Weakly, with the N-Terminal Domain of KPNB1

Given the protective capacity of RTEL1 against importazole-induced effects, we explored the possibility that RTEL1 interacts directly with KPNB1 (KPNB1) using co-immunoprecipitation approaches in unperturbed cells. Antibodies against RTEL1 failed to co-immunoprecipitate KPNB1 or vice-versa when proteins were expressed at endogenous levels (not shown). However, when we overexpressed a tagged version of RTEL1 from an exogenous vector in 293T cells, we could co-immunoprecipitate a minute amount of KPNB1, relative to the levels endogenously expressed in these cells (Appendix A). This result points to a very weak (or else very transient) interaction between RTEL1 and KPNB1 and does not support, without totally ruling it out, the view that RTEL1 physically shields KPNB1 from importazole binding. Nevertheless, we could also specifically co-immunoprecipitate RTEL1 when we overexpressed it in 293T cells along with tagged versions of the N-terminus of KPNB1, which lack the region for interaction with KPNA2 (Appendix A), including a version which no longer interacts with RAN, a key partner for import activity [33] (Appendix A). Further co-immunoprecipitation experiments showed that the site of this interaction in RTEL1 is located on the C-terminus of the protein, between positions 1191 and 1219, which is immediately distal to the PCNA interacting domain in RTEL1 (PIP, aa 1178-1185) (Appendix A). Therefore, these results confirm a physical interaction, albeit transient and/or weak, between RTEL1 and this importin. Unfortunately, and despite all our efforts, we could not pinpoint key RTEL1 residues for this interaction, since all mutations that we tested (and that affected residues chosen on the basis of educated guesses following secondary and tertiary structure predictions) did not abolish the RTEL1/KPNB1 interaction (as evaluated using the overexpression and co-immunoprecipitation approaches described above).

To determine how the RTEL1/KPNB1 interaction might function in cellulo, we overexpressed a GFP fusion of an N-terminal fragment of KPNB1 (here named KPNB1Δ), which interacts with both RTEL1 and RAN but does not carry the site of interactions with KPNA2 (which itself binds to NLS-bearing proteins) [34]. As illustrated in Figure 3D, the fused protein predominantly accumulated in the nucleus, with a reinforcement at the level of the NE. Notably, cells overexpressing the GFP-KPNB1Δ fusion displayed a limited loss of nuclear circularity (Figure 4E). Interestingly, when GFP-KPNB1Δ was expressed in cells overexpressing either RTEL1 WT or NLSmut, GFP-KPNB1Δ was almost exclusively found associated with the NE (Figure 4D), and cells no longer showed nuclear deformations (Figure 4E). On the contrary, when GFP-KPNB1Δ was expressed in cells overexpressing RTEL1-MR, GFP-KPNB1Δ was found both in the nucleus and the cytoplasm, still with some reinforcement at the level of a profoundly distorted NE (Figure 4D,E). Results from these experiments support the link between NE deformations and KPNB1 dysfunction. Moreover, they also support the possibility that RTEL1 ensures NE integrity and stability by mobilizing KPNB1, and either targets it to the nuclear envelope or promotes its passage through the nuclear pore complex (NPC). 

### 3.5. The C-Terminus of RTEL1 Lacking the Helicase Domain Is Sufficient to Protect Cells from Importazole-Induced NE Defects 

To better understand how RTEL1 protects the NE from defects induced by interfering with KPNB1 activities, we determined the minimal regions in the protein that are required to exert the protection against importazole. Overexpression of a C-terminal domain of RTEL1 containing amino acids 683-1300 (therefore excluding the entire helicase domain) (Figure 5A, Appendix A) was able to protect cells from the NE defects induced by importazole (Figure 5B). As observed for the full-length version of RTEL1 mutated in the NLS, the 683-1300 domain carrying mutations in the NLS also protected against importazole-mediated NE defects. These results reveal that the protective effect of RTEL1 against importazole-induced NE defects resides in its C-terminus, is independent from any helicase activity, presumably does not require ATP hydrolysis and does not depend on the classic import pathway. To refine the region in RTEL1 responsible for this protective activity, we further deleted the N-terminus (Figure 5A) and tested the capacity of shorter fragments to protect against importazole. As illustrated in Figure 5B and Appendix A, a fragment carrying domain 781 to 1300 (i.e., still carrying an intact NLS) was unable to protect cells against importazole-induced effects. A similar observation was made with a 763-1300 fragment. These experiments identify region 683-762 in RTEL1 as critical to protect the NE from the deleterious effects of importazole.

### 3.6. The 683-762 Domain in RTEL1 Is Required for the Induction of NE Deformation Phenotypes Caused by Mutations in the RING Domain

We next investigated the role of this importazole protective domain in the spontaneous NE deformations detected in cells carrying mutations in RTEL1. To do this, we tested the capacity of C-terminus fragments of RTEL1 carrying RING mutations to induce NE deformations either in the presence or absence of the 683-762 domain (Figure 5A). As shown in Figure 5C (and Appendix A), only cells released in S-phase while expressing the 683-1300 fragment carrying mutations in the RING displayed pronounced NE deformations, while cells expressing the 763-1300 fragment mutated in the RING did not. This outstanding observation implies that NE destabilizations observed in the presence of RING mutations also require the activities exerted by the 683-762 domain in RTEL1.

### 3.7. A 683-808 Domain in RTEL1 Promotes Nuclear Import through Targeting to the NPC

Given the tight connection between import inhibition and NE deformations and the protective effect exerted by the C-terminus of RTEL1 only in the presence of the 683-762 domain, we explored the possibility that this domain could itself display some pro-import activity. We examined the localization of GFP when fused to the C-terminus of RTEL1 (683-1300) in the presence and absence of mutations in the NLS (Figure 5D, Appendix A). While the fusion GFP-683-1300 predominantly locates in the nucleus, the version carrying mutations in the NLS displays both nuclear and cytoplasmic localizations, confirming the importance of the contribution from the KPNA2/KPNB1-dependent pathway for partial RTEL1 nuclear localization. Interestingly, a fusion of domain 781-1300 to GFP turned out to be totally dependent on an intact NLS for its nuclear localization, since a similar version carrying mutations in the NLS was found almost exclusively in the cytoplasm (Figure 5A,D, Appendix A). These experiments allow us to propose that the 683-780 domain also contributes to the nuclear import of the protein. To directly test whether this domain is indeed endowed with import activity, we selected a fragment spanning amino acids 683 to 808, to consider the presence of prolines in positions 757-766, which could be potentially important for peptide structure. We fused this 683-808 aa domain (therefore excluding the NLS) to GFP and found that the localization of the fused peptide (42.9 kDa) was exclusively nuclear, which contrasted with the only partial nuclear localization for GFP alone (27 kDa) (Figure 5A,D, Appendix A) expected for a protein that is free to passively translocate back and forth through the nuclear pore. This result confirms that the domain in RTEL1 required for both importazole protection and NE instability expressed in the presence of RTEL1 mutations is endowed with a pro-import activity; we therefore refer to this novel domain as an Import Targeting Domain (ITD) (Figure 5A). The import activity of ITD was largely insensitive to importazole, and overexpression of the GFP-ITD fusion protein did not prevent importazole-induced NE deformations (Figure 5E, Appendix A), indicating that, although absolutely necessary, the ITD domain is not sufficient to exert a protective effect against importazole (Appendix A), as it presumably requires a KPNB1 interacting domain and an intact RING domain. 

We next sought to determine if this import-promoting activity could be exerted when fused to a larger protein, as suggested by the fact that GFP-683-1300 (size 95.8 kDa) carrying a mutation in the NLS showed partial nuclear enrichment (Figure 5A–D, Appendix A). We fused the ITD domain to two copies of MBP (maltose-binding protein) in tandem (noted MBPx2-ITD, total size of 101.6 kDa) and checked the subcellular localization of the fusion. As shown in Figure 6A the protein was mostly cytoplasmic, with some reinforcement of the signal in close proximity to the nuclear envelope. This observation indicates that the pro-import activity of 683-808 is not efficient when cargos are larger than the limit imposed by the nuclear pore barrier for free diffusion. Interestingly, staining with our anti-RTEL1 antibody, which recognizes the 683-808 domain, led to the detection of abundant accumulation of the protein next to the nuclear envelope (Figure 6B). Furthermore, these accumulations contained other proteins such as KPNB1, NUP153 and RANBP2 (Figure 6B), suggesting that the fused protein might engage contacts with the nuclear pore to initiate import, but then it is removed from the pore due to non-productive import.

### 3.8. RTEL1 Interacts with NUP153 

Given the accumulation of the MBPx2-ITD fusion protein at the periphery of the nucleus and the colocalization of 683-808-containing aggregates with some nucleoporins, we asked whether RTEL1 itself interacts with some nuclear pore components. In support of this hypothesis was the fact that several nucleoporins had been previously found in mass spectrometry analyses after immunoprecipitation of RTEL1 overexpressed in 293T cells [23]. We set up new experiments and confirmed that anti-RTEL1 antibodies can co-immunoprecipitate NUP153 in cells expressing both proteins at endogenous levels (Figure 7A), even in the presence of importazole. Using GFP fused to truncated versions of NUP153, we mapped the site of interaction with RTEL1 on the N-terminus of the protein (aa 1-434) (Figure 7B). This domain is found upstream to the sites of interaction with KPNB1 and NUP50 (as illustrated for the latter in Appendix A), which are partners of NUP153 known to play important roles in cargo release in the nuclear basket of the NPC [35]. Furthermore, in co-immunoprecipitation experiments, we were able to co-immunoprecipitate the N-terminal fragment of NUP153 (aa 1-434) together with the C-terminal RTEL1 carrying the ITD (683-1300) (Appendix A). Importantly, expression of the ITD tagged with GFP was able to co-immunoprecipitate endogenous NUP153 in addition to NUP62, but not RANBP2 or NUP98 (Appendix A), strongly suggesting that the ITD in RTEL1 may facilitate import by engaging contacts with the central channel of the nuclear pore (NUP62) and the nuclear basket (NUP153). To confirm that RTEL1 itself can be found at the level of the nuclear pore, we used high-resolution Structural Illumination Microscopy (SIM) to simultaneously visualize endogenous RTEL1, one structural nuclear pore component and the nuclear envelope in unperturbed HeLa cells. As illustrated in Figure 7C, RTEL1 can be observed near the cytoplasmic face of the nuclear pore, in very close proximity to RANBP2, as well as on the nuclear face of the nuclear pore.

## 4. Discussion

In this study, we have shown that RTEL1 deficiency is directly associated with S-phase specific nuclear envelope (NE) disruptions. We also have shown that RTEL1 nuclear association is stimulated in S-phase, and that NE destabilizations induced by RTEL1 mutations require origin firing. Interestingly, blocking major protein import pathways using a KPNB1-specific inhibitor also led to NE destabilizations, albeit not related to the cell cycle. This effect was totally absent in cells overexpressing wild-type RTEL1. We were able to show that RTEL1 interacts both with KPNB1 and NUP153, the latter being a key factor for cargo release in the basket of the nuclear pore complex and the nuclear pore protein with the highest affinity for KPNB1 [25,36]. In addition, using high-resolution microscopy, we showed that RTEL1 is present on both sides of the NE in close contact with nuclear pore components. Finally, we showed that the NUP153-interacting domain in RTEL1 promotes nuclear internalization of peptides not larger than the size limit imposed by the nuclear pore complex. In all, this study further highlights the multifunctionality of RTEL1 and reveals hitherto unknown links between its S-phase functions and its role in NCT. 

The nuclear deformations associated with RTEL1 deficiency are characterized by local depletions in lamin B1, particularly at the level of pronounced nuclear blebs. We have demonstrated through multiple lines of experimental evidence, using the RTEL1 RING domain as a model for mutations found in humans, that such phenotypes are a direct consequence of RTEL1 dysfunction and only become detectable when cells enter S-phase. The fact that the absence of the protein and the expression of a mutated form both lead to the same consequences at the level of the NE indicates that RTEL1 is directly responsible in the bridging of S-phase to NE integrity. The roles of RTEL1 in S-phase appear to be multiple and are related to the progression of the replication fork and to telomere lengthening. However, although the NE manifestations described here do require origin firing, they do not require DNA synthesis, suggesting that this NE-related RTEL1 function takes place between the moment the MCM helicase is activated and the formation of an active polymerization complex behind the MCM ring. The precise nature of this function remains to be defined. Nevertheless, the work presented here strongly suggests that this function is related to a nuclear import process that involves KPNB1, as we have identified a novel interacting domain in RTEL1 for KPNB1, and that this domain is required for the protection of cells against treatment with importazole, a specific inhibitor for KPNB1. Although RTEL1 itself appears to be imported to the nucleus more efficiently in S-phase, preventing this enrichment (through mutations in the NLS) was not sufficient to induce an overt NE phenotype. Although this conclusion is drawn from experiments where the protein bearing the mutated NLS was barely overexpressed, they still leave open the possibility that the absolute number of RTEL1 molecules that reach the nucleus despite the mutation is sufficient to carry out the required function, as all other domains remain intact. It should be noted that, in unperturbed HeLa cells, the number of RTEL1 molecules has been calculated to be 5000-fold less than that of KPNB1 [37], which is one of the most abundant proteins in the cell. 

Our unprecedented observation that perturbing nuclear import using a specific KPNB1 inhibitor is sufficient to cause NE abnormalities supports the notion of a tight link between import activities and NE stability. However, the NE defects seen after inhibiting KPNB1 lack the cell cycle specificity shown by RTEL1 dysfunctions, strengthening the concept of an S-phase-specific import-related phenomenon that directly involves RTEL1 function. Notably, we have demonstrated that RTEL1 can mobilize an NE-associated KPNB1 that has lost its ability to interact with KPNA2, and that this activity requires wild-type RTEL1, further supporting a critical role for RTEL1 in KPNB1-dependent import. Although our experimental data indicates that RTEL1 physically interacts with both KPNB1 and NUP153, we have no indication that these interactions are cell cycle-regulated, simultaneous or even perturbed by the same mutations in RTEL1 that induce NE deformations. Certainly, experiments where RTEL1 (either the whole protein or the import-relevant fragments identified here) is modified with a biotin ligase domain [38] could be very informative to either confirm these interactions (including their cell cycle specificity) or identify new ones. Notwithstanding, our work remarkably shows that RTEL1 can prevent the consequences of KPNB1 inhibition, that this quality resides in its C-terminus and that it does not involve the helicase domain. 

In support of our proposed activities for RTEL1 in import, we identified a specific domain (ITD) between positions 683-808 in RTEL1, able to both stimulate the association of proteins with nuclear pore components and promote the nuclear translocation of small peptides in a KPNB1-independent (and presumably RANGTP-independent) way. Intriguingly, this ITD domain, which interacts with NUP153, is absolutely required (albeit not sufficient) to protect cells from the consequences of KPNB1 inhibition by importazole. Although the pro-import activity of ITD does not seem to have a cell cycle preference, it is possible that, when in the context of the full-length protein, it contributes to the nuclear localization of RTEL1 itself by bringing RTEL1 into proximity of the NPC and KPNB1 (Figure 8). 

Whether other proteins that need to be enriched in the nucleus in S-phase can use the same import pathway remains a possibility, even if we could not find similar domains in the protein database. It is important to note that, although nuclear import has been studied for decades, cargoes for specific transporters have only recently begun to be elucidated [39,40,41]. Nevertheless, in the context of S-phase, the precise and timely regulation of nuclear transport is imposed by the fact that numerous import and export cargos can populate the NPC at any given time [26], a phenomenon that probably peaks at the entry into S-phase, when rDNA transcription, ribosomal subunit production and histone translation are increased [42,43]. Specifically, the nuclear enrichment of RTEL1 via a mechanism involving both its ITD and KPNB1 could be essential to the function exerted by the former. Another, perhaps more intriguing, possibility is that RTEL1, through its ITD, might directly stimulate the KPNB1-mediated import of other factors required for correct S-phase progression. Moreover, the apparent pleiotropy associated with RTEL1 deficiencies may be a consequence of secondary perturbations in the way genome maintenance-related factors relate to the NPC (and in particular to NUP153) to exert their function [44,45]. Once again, Bio-ID experiments [38] could provide interesting answers to these questions. In all, our results suggest that mutated RTEL1, or the absence thereof, can perturb the KPNB1-dependent import process (of itself or of other proteins). This effect might result in reduced functionality of the NPC, perhaps including blockage, permeability alterations or even aberrant directionality of transport [46], all of which could lead to remodeling or disassembly of the NPC. How this NPC response is connected to origin firing in S-phase and how it leads to nuclear envelope destabilization remain to be elucidated. Another remaining question is whether import/nuclear envelope-related RTEL1 activities are conserved throughout species, particularly in mouse RTEL1, the most-used mouse model for RTEL1 studies. Interestingly, amino acid sequence alignments between mouse and human RTEL1 at the level of ITD or the KPNB1-interacting domain reveal substantial divergences (while remaining very well conserved in hominin), suggesting a recently acquired, helicase-independent, function for this essential protein (Appendix A). 

Whatever the case, our results suggest that perturbations in these novel functions of RTEL1 may contribute to the clinical manifestations seen in patients carrying mutations in the gene. One prominent biological manifestation of RTEL1-associated diseases is cellular senescence [47,48], frequently attributed to the presence of short telomeres in these cells. Nevertheless, the results presented here open the possibility that senescence may have other causes. Indeed, disruption of NCT functions is a senescence driver [49] and aging itself may have deleterious effects on NPC composition and function, leading to a loss of the nuclear permeability barrier and the leaking of cytoplasmic proteins into the nuclear compartment [50]. Of note, it has been shown that, in the context of progeria (due to LMNA/C mutations), NE destabilizations are associated with perturbed nuclear transport, essentially explained by alterations in the composition of NPC [51]. Interestingly, restoring NPC integrity (by allowing the mobilization of importins and nucleoporins trapped in the cytoplasm) corrected cellular phenotypes in vitro [51] and enhanced the lifespan of mice carrying LMNA/C mutations [52]. These observations, plus the fact that LMNB1 loss has been identified as a senescence marker [53], reinforce the idea that the NCT perturbations and the NE destabilizations we report here may contribute to cell senescence in patients carrying RTEL1 mutations. 

## 5. Conclusions

The work presented here offers a specific and unexpected perspective on the mechanism of RTEL1 dysfunction. The identification of new pathological processes at the cellular level, and directly connected to RTEL1 mutations, expands our understanding of the nature of RTEL1-related diseases and hence, possible avenues of treatment.

## Figures and Tables

**Figure 1 cells-12-02798-f001:**
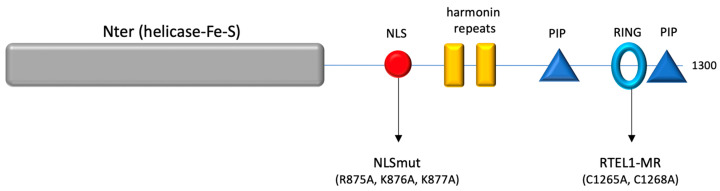
Diagram representing functional domains in the RTEL1 protein. Indicated are the helicase domain in the N-terminus, the location of the nuclear localization signal (NLS, with the positions mutated in the NLSmut version of the protein), the harmonin repeats and the PIP boxes bracketing the RING domain, where the positions indicated were mutated in the MR version of the protein.

**Figure 2 cells-12-02798-f002:**
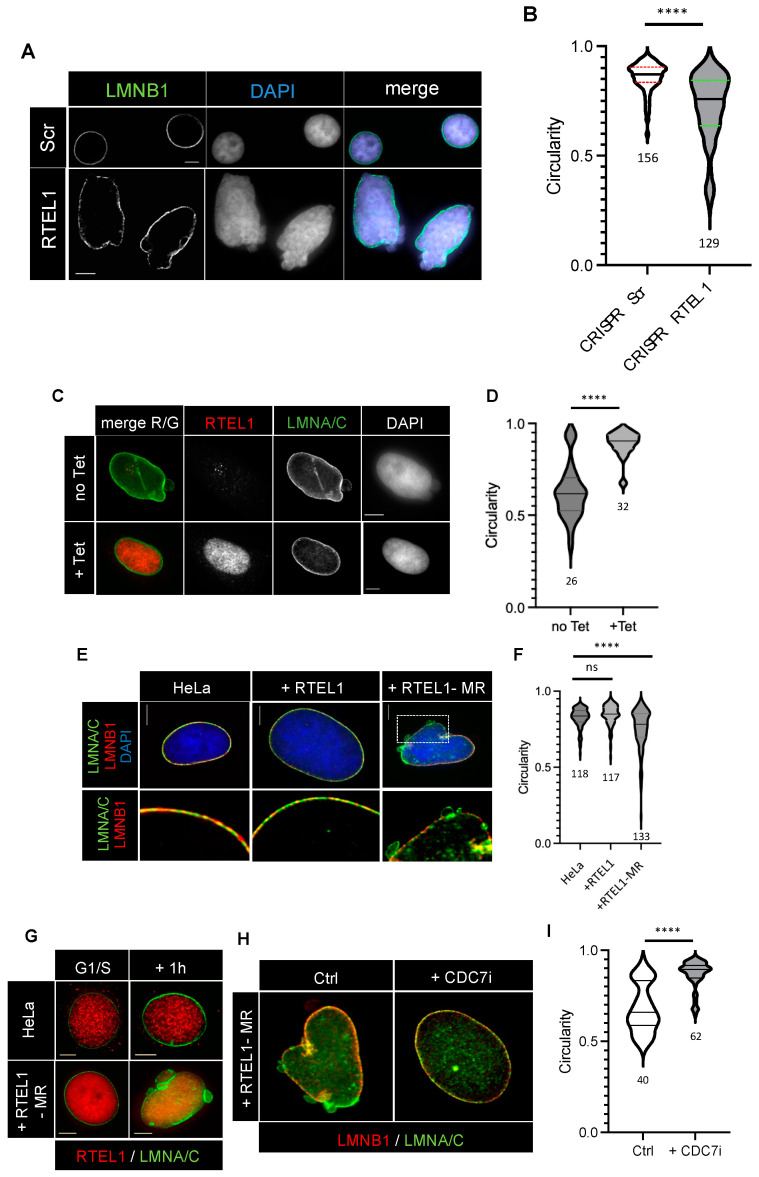
(**A**,**B**) HeLa cells transfected with a CRISPR-Cas9-IRES-GFP construction targeting human RTEL1 for 48 h were cell sorted and plated on slides and grown for further 18 h; cells were fixed in PFA and indirect immunofluorescence was performed using anti-LMNB1 antibodies, and they were counterstained with DAPI. Circularity of individual nuclei was measured using ImageJ. (**C**,**D**) A CRISPR-Cas9-IRES-GFP treatment was performed as just described using HeLa cells carrying a stable Tet-inducible RTEL1 transgene. After FACS sorting, plated cells were grown in the presence of Tet for 18 h and then indirect immunofluorescence was performed using anti-LMNA/C and anti-RTEL1 antibodies, and they were counterstained with DAPI before circularity of nuclei was measured. (**E**,**F**) Unperturbed HeLa cells or cells stably carrying Tet-inducible forms of RTEL1 transgenes (either WT or with mutations in the RING domain -MR) were synchronized at G1/S via a double thymidine block. Tet was added to the cells during the second thymidine block and then cells were released into S-phase for 1 h, fixed and treated for indirect immunofluorescence with anti-RTEL1 and anti-LMNA/C antibodies before circularity of nuclei was measured. (**G**) Synchronized HeLa cells expressing (or not) RTEL1-MR were subjected to a 2x thymidine block as described above, then released into S-phase in the presence of aphidicolin (1 μg/mL), fixed and treated for indirect immunofluorescence stained with anti-RTEL1 and anti-LMNA/C antibodies. (**H**,**I**) Synchronized HeLa cells expressing RTEL1-MR were treated with the CDC7 inhibitor PHA76749 (5 μM) before release into S-phase; cells were treated for IF using anti-LMNA/C and anti-LMNB1 before circularity was measured. (**A**,**C**,**E**,**G**,**H**) Scale bars are 5 μm. (**B**,**D**,**F**,**I**) All statistical comparisons were performed using a Mann–Whitney test (**** = *p* < 0.0001, ns: not significant). The number of cells analyzed is indicated.

**Figure 3 cells-12-02798-f003:**
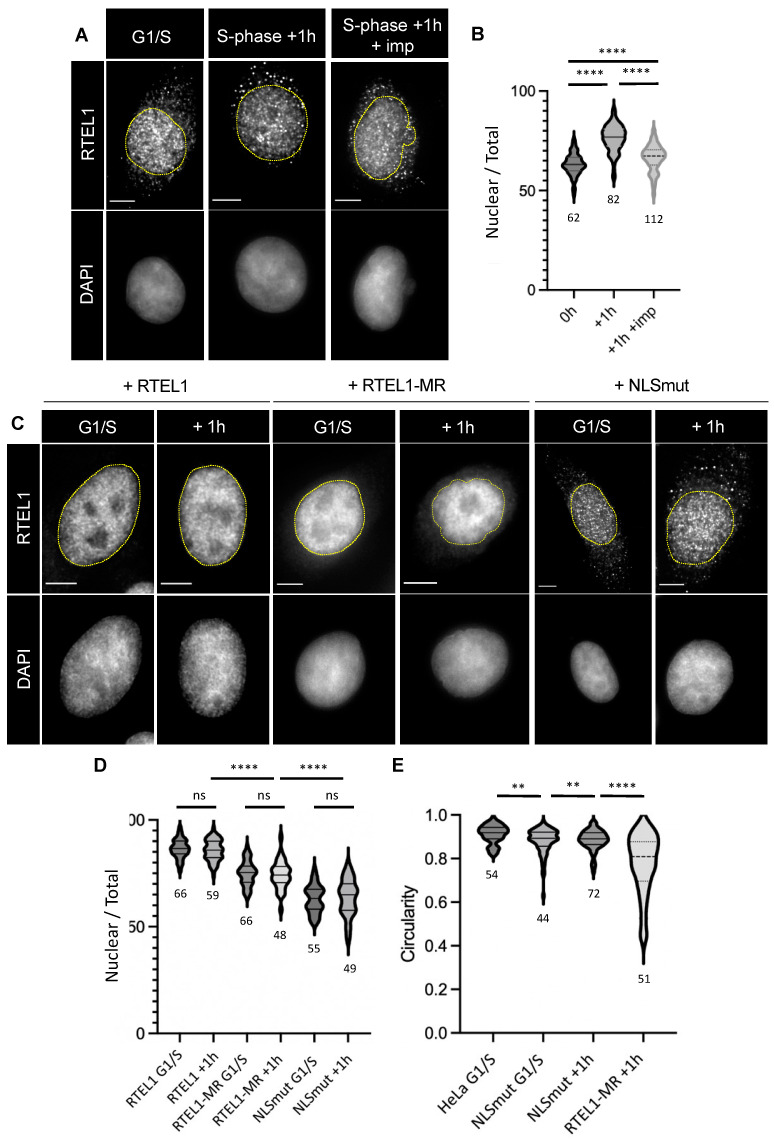
(**A**,**B**) HeLa cells were subjected to a double thymidine block and released into S-phase for 1h in the presence (or not) of importazole (40 μM for 3 h). Cells were fixed for IF with anti-RTEL1 antibodies and counterstained with DAPI. Fluorescence intensities in nuclear and cytoplasmic compartments were measured using ImageJ, with masking on the DAPI signal. The fraction of nuclear signal (Nuclear/Total) was calculated with the formula = nuclear intensity-background intensity/total intensity-background intensity. (**C**,**D**) Synchronized HeLa cells induced for the expression of either wild-type RTEL1, RTEL1-MR or NLSmut were blocked at G1/S or released in S-phase for 1 h. Cells were fixed for IF with anti-RTEL1 antibodies and counterstained with DAPI. The fraction of nuclear signal was calculated as in (**A**). (**E**) Circularity of nuclei was measured as above. (**A**,**C**) Scale bars are 5 μm. (**B**,**D**,**E**) Statistical comparisons were performed using a Mann–Whitney test (** = *p* < 0.001; **** = *p* < 0.0001), and the number of cells analyzed is indicated.

**Figure 4 cells-12-02798-f004:**
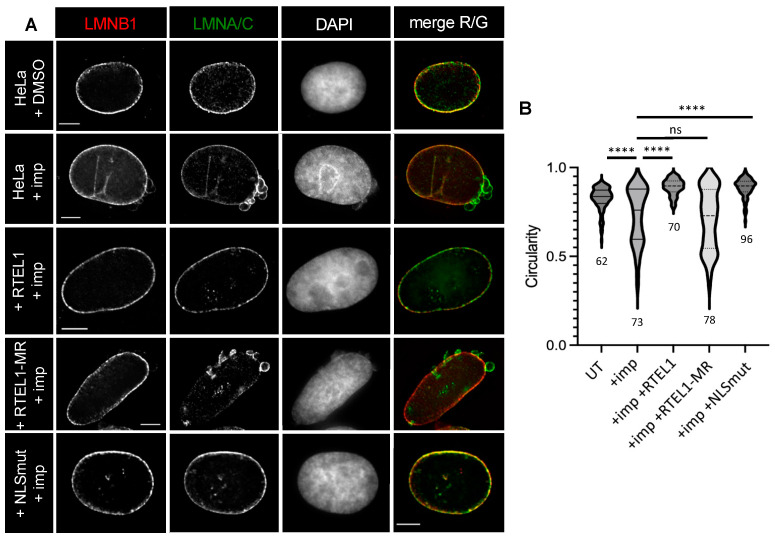
(**A**,**B**) Synchronized HeLa cells either uninduced or induced for the expression of either wild-type RTEL1, RTEL1-MR or NLSmut were blocked at G1/S and treated with importazole for 3 h (40 μM); HeLa cells blocked at G1/S were also treated with DMSO as a control. Cells were fixed for IF with anti-lamin antibodies (LMNA/C and LMNB1) and counterstained with DAPI before circularity of nuclei was measured. (**C**) Synchronized HeLa cells induced for the expression of either wild-type RTEL1, RTEL1-MR or NLSmut were blocked at G1/S and treated for 3 h with ivermectin (IVM, 50 μM for 3 h). Cells were fixed for IF with anti-RTEL1 antibodies and counterstained with DAPI. (**D**,**E**) After transfection with a vector expressing a GFP-tagged C-terminal truncation of KPNB1 (unable to bind KPNA2), HeLa cells either uninduced or induced for the expression of either wild-type RTEL1, RTEL1-MR or RTEL1-NLSmut were blocked at G1/S and treated for 3 h with importazole (40 μM for 3 h). Cells were fixed for IF with anti-LMNA/C and counterstained with DAPI before circularity of nuclei was measured. GFP signal was directly visualized on the green channel. LMNA/C-rich NE bleb is indicated by the arrow and enlarged right below. (**A**–**D**) Scale bars are 5 μm (except in the inset, where it represents 2 μm). (**B**,**E**) Statistical comparisons were performed using a Mann–Whitney test (**** = *p* < 0.0001, ns: not significant), and the number of cells analyzed is indicated.

**Figure 5 cells-12-02798-f005:**
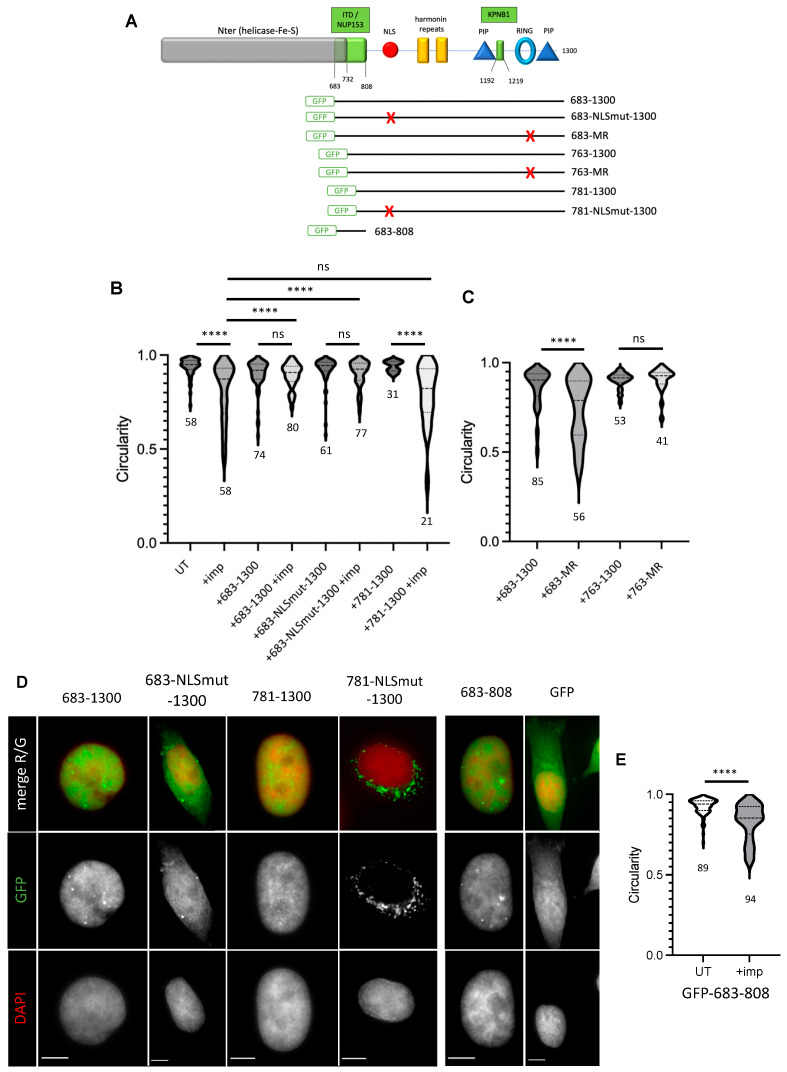
(**A**) Representation of RTEL1 indicating known functional domains and those identified in this study (in green). The fragments fused to GFP at the N-terminus are indicated. Mutations in the NLS or the RING domain, as previously described, are indicated as a red X. (**B**) Synchronized HeLa cells (see Appendix A) induced (or not) for the expression of RTEL1 C-terminal fragments (the most N-terminal position for each construction is indicated) lacking the helicase domain and fused to GFP were blocked at G1/S and treated with importazole for 3 h (40 μM). The 683-NLSmut-1300 carries mutations abolishing the classic import pathway. Cells were fixed before IF with anti-LMNA/C antibodies and counterstained with DAPI before circularity of nuclei was measured. (**C**) Synchronized HeLa cells (see Appendix A) induced (or not) for the expression of RTEL1 C-terminal fragments (positions are indicated) lacking the helicase domain, fused to GFP and carrying (or not) mutations in the RING domain were blocked at G1/S, and then released into S-phase for 1 h. Cells were fixed, treated for IF with anti-LMNA/C antibodies and counterstained with DAPI before circularity was measured. (**D**) Synchronized HeLa cells induced (or not) for the expression of GFP fusions of RTEL1 C-terminal fragments (positions are indicated), lacking the helicase domain and carrying mutations in the NLS (when indicated NLSmut), GFP fused to the ITD (683-808) or GFP alone, were blocked at G1/S, and then released into S-phase for 1 h. Cells were fixed, treated for IF with anti-LMNA/C antibodies and counterstained with DAPI before examination under the microscope. GFP signal was directly visualized on the green channel and its intensity measured to calculate nuclear enrichment as above (see Appendix A). Scale bar is 5 μm. (**E**) Synchronized HeLa cells (Appendix A) induced for the expression of the GFP fused to the 683-808 domain of RTEL1 were blocked at G1/S and then treated with importazole for 3 h (40 μM). Cells were fixed, treated for IF with anti-LMNA/C antibodies and counterstained with DAPI before circularity was measured. (**B**,**C**,**E**) Statistical comparisons were performed using a Mann–Whitney test (**** = *p* < 0.0001), and the number of cells analyzed is indicated.

**Figure 6 cells-12-02798-f006:**
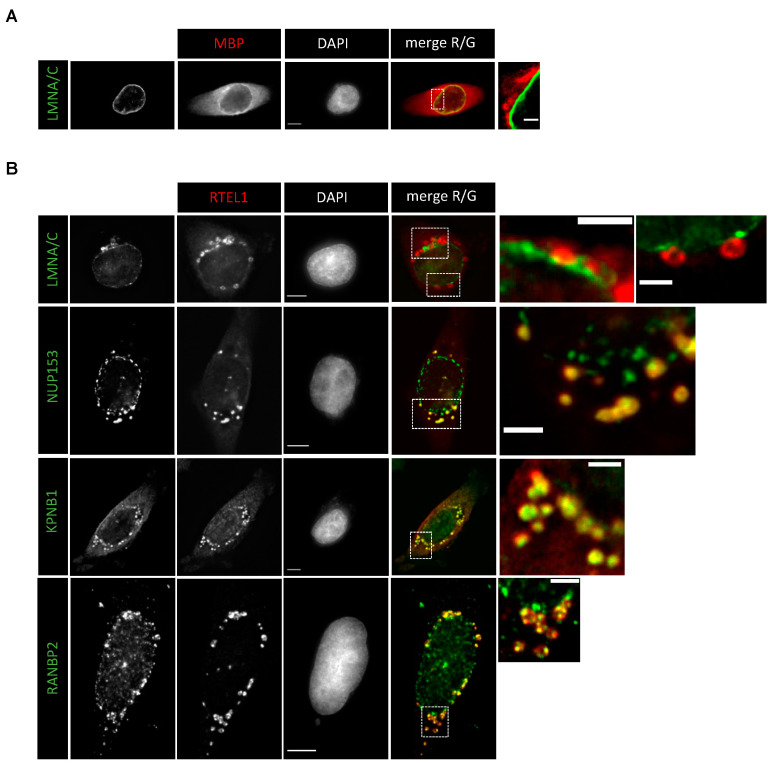
(**A**) HeLa cells were transiently transfected with 2x-Maltose-binding protein (2xMBP) fused to RTEL1-ITD (683-808); 24h later, cells were fixed and treated for IF using anti-MBP and counterstained with DAPI. (**B**) HeLa cells were transiently transfected with 2x-Maltose-binding protein (2xMBP) fused to RTEL1-ITD (683-808); 24 h later, cells were fixed and treated for IF using anti-RTEL1 in combination with either anti-LMNA/C, anti-NUP153, anti-KPNB1 or anti-RANBP2 and counterstained with DAPI. Of note, the anti-RTEL1 antibody recognizes the ITD domain. (**A**,**B**) Scale bars are 5 μm except in insets, where it represents 2 μm.

**Figure 7 cells-12-02798-f007:**
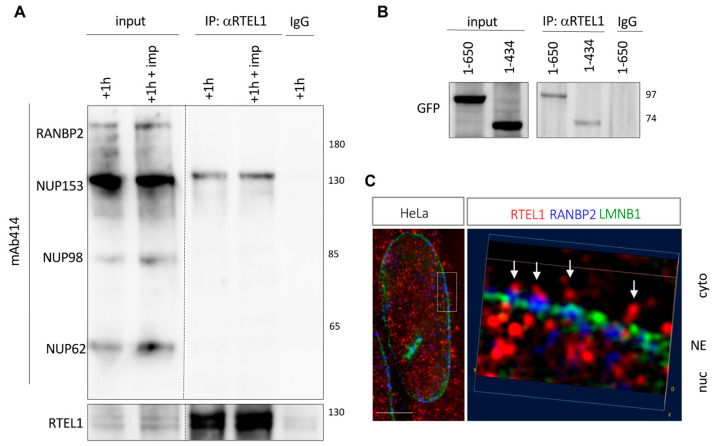
(**A**) Hela cells were synchronized in G1/S, treated (or not) with importazole for 3 h (40 μM) and then released into S-phase for 1 h. Lysates were prepared and treated for immunoprecipitation with anti-RTEL1. Immunoprecipitates and inputs were analyzed via Western blot using a monoclonal anti-mAb414 antibody, recognizing the FG repeats present in the indicated Nuclear Pore Complex components, as well as anti-RTEL1. (**B**) 293T cells were transiently transfected with GFP-tagged N-terminal fragments of NUP153 (positions indicated), and lysates were treated for immunoprecipitation with anti-RTEL1 antibodies. Immunoprecipitates were analyzed via Western blot using an anti-GFP antibody. (**C**) Super-resolution microscopy (SIM) image of an unperturbed Hela cell fixed with PFA and treated for IF using anti-RTEL1, anti-RANBP2, and anti-LMNB1 antibodies. The square on the left indicates the section of the nuclear envelope that was enlarged on the right. Nuclear pore complexes are indicated by arrows. NE: Nuclear envelope, Cyto: cytoplasmic face, Nuc: nuclear face. Scale bar is 5 μm**.** (**A**,**B**) Molecular weights in kDa are indicated.

**Figure 8 cells-12-02798-f008:**
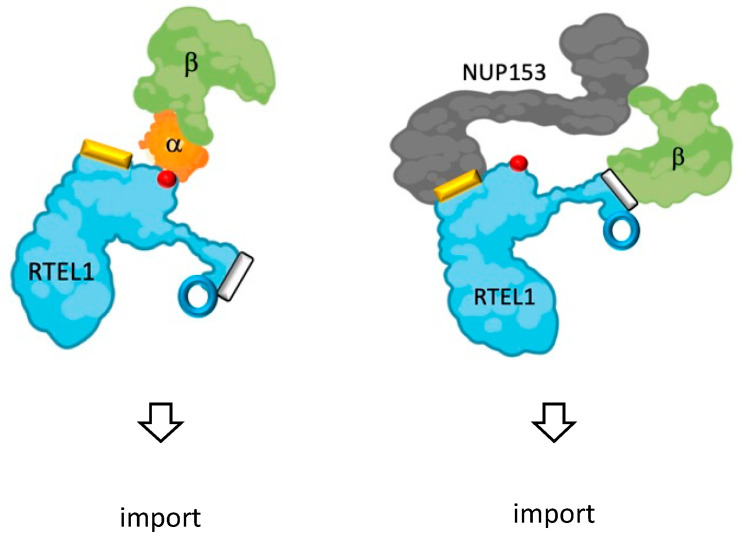
Proposed model for RTEL1’s modes of import: On the left, classic import pathway in which the NLS (closed red circle) of RTEL1 mediates the interaction with KPNA2 (importin α), which in turn is bound by KPNB1 (importin β) to be imported. On the right, RTEL1 directly interacts both with NUP153 (which itself interacts with KPNB1) through the ITD (yellow rectangle) and with KPNB1, through a region (white rectangle) that is close to the RING (open blue circle). It is possible that the latter interaction only occurs in the nuclear basket after the RTEL1-KPNB1 complex travels through the nuclear pore with the help of the ITD. Of interest, mutations in the RING render the RTEL1 protein highly dependent on KPNA2 for nuclear import, indicating that the KPNB1-NUP153 pathway is no longer active. Both the mechanisms that restrict the latter to S-phase (perhaps post-translational modifications of the ITD?) and how mutations in RTEL1 lead to nuclear envelope disassembly remain to be discovered.

## Data Availability

Most data are contained within the article or Appendix A. Original uncropped images and original Prism files carrying quantification results and used to generate violin plots are available from the corresponding author upon request.

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
