# Peer review of "Human RTEL1 Interacts with KPNB1 (Importin β) and NUP153 and Connects Nuclear Import to Nuclear Envelope Stability in S-Phase"

_cells, 2023, doi:10.3390/cells12242798_

Round 1

Reviewer 1 Report

Comments and Suggestions for Authors

This is an interesting work points to elucidate the helicase-independent role of RTEL1 in connecting both nucleo-cytoplasmic trafficking and nuclear envelope integrity to genome replication in S-phase.

However, I have some suggestions which may improve the quality of the paper and readability.

Major Points:

-        Rows 167-168: ‘RTEL1 carrying engineered mutations that mimic those of HHS patients (Figure 1E-F), in this case mutations affecting the RING domain (hereafter named MR, for Mutated Ring)’. It would be very useful to see a schematic representation of the protein and its mutated domains.

-        Rows 222-224: ‘As illustrated in Figure 2A-B, endogenous RTEL1 shows a preferential nuclear localization (slightly above 60% following this 223 approach) in G1/S, which significantly increases (reaching almost 80%) upon entry into S-phase.’ I have some concerns about that. First, I would suggest including the DAPI staining. Second, a more elegant experiment to conclude a nuclear localization of RTEL1 could be given by chromatin fractionation Assay (see suggested references below). Moreover, how authors can be sure that cell synchronization has properly worked? I suggest including appropriate control (WB or FACS). I strongly suggest referring to the following works for both points:

o   Messina G, et al. The ATPase SRCAP is associated with the mitotic apparatus, uncovering novel molecular aspects of Floating-Harbor syndrome. BMC Biol. 2021 Sep 2;19(1):184. doi: 10.1186/s12915-021-01109-x.

o   Messina G, Atterrato MT, Prozzillo Y, Piacentini L, Losada A, Dimitri P. The human Cranio Facial Development Protein 1 (Cfdp1) gene encodes a protein required for the maintenance of higher-order chromatin organization. Sci Rep. 2017 Apr 3;7:45022. doi: 10.1038/srep45022.

-        Fig.2A-B: It is not clear if ‘NLS’ is the wt or the mutated one. I presume NLSmut. If it is true, why authors did not include the wt NLS as control? I suggest doing that.

-        Fig.3A: I would also include here a picture of HeLa treated with importazole diluent (DMSO, I suppose) as negative/positive control.

Minor Points:

-        Rows 325-327: ‘This result points to a very weak (or else very transient) interaction between RTEL1 and KPNB1 and does not support, without totally ruling it out, the view that RTEL1 physically shields KPNB1 from importazole binding’. This is true. I suggest authors to take in consideration the BioID assay for the future of the project. Maybe it can be discussed in ‘Discussion’ or ‘Future Perspectives’ paragraphs.

-        Row 335: I would suggest relocating the schematic representation of the truncated constructs and mutation points to a main figure rather than hide it in the supplementary figures. I believe that it would be really appreciated by readers who seek precise comprehension of the protein versions being analysed. Authors could integrate this information with what was mentioned in lines 167-168.

-        Row 342: ‘in cellulo’ should be changed in italic.

-        Figure 5: Authors should include the magnitude rate of the pictures to the right.

-        The MW of all the WB bands and specific used ladder should be included in all the figures (supplementary and unpublished included).

-        Row 502-503: ‘We have shown here that RTEL1 mutations in HHS or DC patients are associated with nuclear envelope deformations.’ I was wondering if these phenotypes are somehow related to aging or progeria. Authors could include few comments about that in the discussion paragraph.

Author Response

See file attached

Reviewer 2 Report

Comments and Suggestions for Authors

In their manuscript entitled „ Human RTEL1 interacts with importin β and NUP153 and connects nuclear import to nuclear envelope stability in S-phase”, Schertzer and colleagues demonstrate a function of RTEL1 in maintaining nuclear envelope stability which is connected to nuclear import. Through many experiments, the authors build up a model in which way RTEL1 could be involved in this process and characterize the potential regions within RTEL1 that are required for nuclear envelope stability and nuclear import, respectively.

This is an interesting and well-conducted work, and the results are displayed in a clearly arranged and logical way. However, the amount of experiments performed and number of supplementary figures made the reading of the results part difficult.

I recommend the publication of this article, but suggest that the discussion part repeats the main findings in a short summary. The proposed model displayed in Fig. S6A should be included into the main paper and discussed/explained in the summary to ensure that the reader is able to follow the sometimes complicated findings.

Minor Comments:

1.     Concerning the quantification of cells: How many cells have usually been measured?

2.     Fig. S1 A-C: What is meant by P2 and P25 (patients number? Passage of the cells?)

3.     Fig. S2A: Why does the overexpression of RTEL1 not lead to an increase of the signal (compared to the first column, where cells are not overexpressed)?

4.     Line 332 “Further co-immunoprecipitation experiments showed that the site of this interaction in RTEL1 is located on the C-terminus of the protein, between positions 1191 and 1219…” but shown is the construct 1164, not 1191

5.     Fig. 3D: Please change to KPNB1Δ, so that it fits to the text.

6.     Is importin β synonymous with KPNB1? If yes, I would avoid the switching of nomenclature and decide for only one version.

Reviewer 3 Report

Comments and Suggestions for Authors

This manuscript describes that RTEL1 plays a role in regulating DNA replication through interaction with importin and NUP153. I have no serious concerns, but only have some specific comments to improve readability.

1. Lines 47-62: The last paragraph of Introduction describes too much of the results redundant with Abstract and Results sections. It is recommended that the last paragraph describes what remains unsolved and then describes briefly what or how it has been solved in this study in a sentence or two. The preceding paragraphs may be expanded to describe more background about potential questions on RTEL1 in relation with NE, nuclear import, the cell cycle, DNA replication, and so on.

2. It would be helpful to provide a diagram of domain structures of RTEL1 in Figure 2: helicase domain, NLS domain, MR domain, the 683-762 domain, 683-808 domain, 683-780 domain, PCNA interacting domain, etc.

3. Lines 83 and 168: “RTEL1 carrying mutations in the RING”. It is recommended to define the mutant RTEL1-MR. Several different terms are used for RTEL1-MR mutants. RTEL1-MR mutant carries both C1265A and C1268A or two versions of mutant exist? Please clarify this.

4. Lines 153: It says “heterozygous staining with antibodies against LMNB1 and LMNA/C and the presence of blebs depleted for LMNB1” but LMNA/C staining is missing in Suppl Fig. 1B. Suppl Fig. 1A and B also shows P2 and P25. Do they stand for passage 2 and 25? This is not described anywhere in the legend or Materials and Methods.

5. Line 175: It would be easier to read for readers if it is described “cells blocked in G1/S with double thymidine block” at this first appearance.

6. Line 323: “a minute amount of KPNB1”. It is not obvious which band is KPNB1 in Supplementary Fig. 3D. What does the asterisk indicate in Supplementary Fig. 3D and E?

7. It would be helpful to provide a summary of functional domains, such as ITD, of RTEL1 as a diagram.

Round 2

Reviewer 1 Report

Comments and Suggestions for Authors

Thanks to the authors for significantly improving the manuscript.